# DyCodeExplainer: Explainable Dynamic Graph Attention for Multi-Agent Reinforcement Learning in Collaborative Coding

## Abstract

We propose **DyCodeExplainer**, a novel multi-agent reinforcement learning (MARL) framework that integrates dynamic graph attention with explainability techniques to improve collaborative coding. Existing MARL systems typically depend on static communication protocols which are not flexible and transparent in performing coding tasks that are more complicated. The above method suffers from this limitation by treating the interaction of agents in the form of a time-evolving graph in which the nodes represent coding agents, and edges indicate messages exchanged between them. A dynamic graph attention network (DGAT) dynamically prioritizes the messages considering contextually relevant message, whereas hard attention gate eliminates noises and helps improve decision-making efficiency. Furthermore, the framework includes gradient-based attention attribution and rule-based post-hoc explanations to explain message prioritization for providing interpretable budgetary information about the collaborative process. The policy and critic networks use Transformer-XL and graph neural networks respectively for managing the long-range dependencies and assessing the memory argument of the joint state values. Experiments show DyCodeExplainer to be more accurate in terms of code correctness and collaborative efficiency than traditional MARL baselines. The novelty of the system is the simultaneous optimization of thresholds for dynamic attention and explainability rules to bridge an important gap in transparent multi-agent coding systems. This work will move the field forward by providing a scalable and interpretable solution for collaborative software development.

## 1 Introduction

Collaborative coding environments (multiple agents working in concert with one another to develop and debug software) pose unique challenges in message prioritization and in decision-making transparency. Traditional multi-agent reinforcement learning (MARL) approaches often treat communication as a static process, failing to adapt to the dynamic nature of coding tasks where message relevance evolves with context (Busoniu et al., 2006). While graph-based MARL methods have shown promise in modeling agent interactions (Corso et al., 2024), they lack mechanisms to explain why certain messages (e.g., error reports or code suggestions) receive higher priority during collaboration. This opacity makes it difficult to trust and to actually use in software engineering workflows.

Recent advances in dynamic graph attention networks (Brody et al., 2021) and explainable AI (Wang et al., 2024) offer potential solutions but remain underexplored in collaborative coding scenarios. Existing systems either focus on heuristic message prioritization (Apathy et al., 2024) or treat explainability as an afterthought, resulting in suboptimal trade-offs between performance and interpretability. For instance, while gradient-based attribution methods can identify important messages, they often produce unstable explanations for sparse, high-dimensional coding inputs (Tan, 1993). Rule-based post-hoc approaches (Vilone & Longo, 2021) mitigate this issue but struggle to generalize across diverse coding phases (e.g., syntax checking versus performance optimization).

We introduce **DyCodeExplainer**, a MARL framework that unifies dynamic graph attention with hybrid explainability techniques for collaborative coding. The coding process is represented by our system in terms of a temporal graph such that nodes correspond to agents (developer, static analyser, etc.) and edges correspond to message exchanges (code patches, warnings, etc.). A gated dynamic attention mechanism is used to adaptively weigh the messages based on their contextual relevancy using hard attention to filter out noise and soft attention to distribute focus of critical inputs. Unlike prior work (Foerster et al., 2016), our approach jointly optimizes attention thresholds and explanation fidelity through:

1) **Adaptive sparsification**: Hard attention gates prune low-weight edges, ensuring only salient messages influence agent decisions while maintaining differentiable paths for gradient-based attribution.

2) **Dual-phase explanations**: Gradient scores from the critic network identify globally important messages, while rule-based mappings (e.g., "prioritize type errors during compilation") provide human-readable justifications for local decisions.

This combination acts against some major limitations of existing methods. First, the dynamic graph representation captures evolving agent roles and dependencies, outperforming static communication protocols (Albrecht et al., 2024). Second, the hybrid explainability framework bridges the semantic gap between low-level attention weights and high-level coding logic, a challenge noted in (Yu et al., 2024). Experiments show a 23% improvement in code correctness over MARL baselines while reducing explanation entropy by 41%.

The main contributions of this work are the following:

- A **dynamic graph attention** mechanism that adjusts message prioritization based on coding phase and agent expertise, using gated sparsification to balance focus and computational efficiency.

- A **hybrid explainability** framework integrating gradient-based attribution with domain-specific rules, enabling both precise importance scoring and intuitive rationale generation.

- Empirical validation showing that DyCodeExplainer enhances both performance (task completion time, code quality) and interpretability (explanation consistency, user trust) in collaborative coding tasks.

The rest of this paper is organised as follows: Section 2 reviews related works in MARL and explainable AI for collaborative systems. Section 3 provides a mathematical formulation of dynamic graph attention and explainability techniques. Section 4 outlines the DyCodeExplainer architecture and this is followed by experimental results presented in Section 5. We discuss some implications and future directions in Section 6 before ending in Section 7.

## 2 RELATED WORK

The development of multi-agent reinforcement learning (MARL) systems through collaborative tasks has seen great strides, especially in fields that require collaborative activities to have the capacity for dynamic purpose interaction and communication. Prior works can be broadly divided into three categories: (1) MARL for cooperative tasks, (2) dynamic graph-based attention mechanisms, and (3) explainability in the multi-agent systems.

### 2.1 MULTI-AGENT REINFORCEMENT LEARNING FOR COOPERATIVE TASKS

Early MARL approaches often treated agents as independent learners, ignoring the benefits of structured communication (Tan, 1993). Later works introduced centralized training with decentralized execution (CTDE) to improve coordination, as seen in methods like QMIX (Rashid et al., 2020). However, these techniques have difficulties with scale when it comes to tasks where a fine-grained message passing is required (e.g., collaborative coding). Recent efforts, such as (Yu et al., 2024), explored conversational interfaces for agent coordination but lacked mechanisms to adaptively prioritize messages based on task context.

## 2.2 Dynamic Graph Attention in Multi-Agent Systems

Graph neural networks (GNNs) have emerged as a powerful tool for modeling agent interactions, particularly in dynamic environments. Wang et al. (2023) proposed a dynamic GNN with sparse attention to improve interpretability, but their method was not designed for sequential decision-making tasks like coding. Similarly, Salehi et al. (2025) introduced prioritization mechanisms for network slicing, demonstrating the benefits of task-aware attention. However, their approach was not concerned with the temporal evolution of communication graphs which we know is also critical in collaborative coding where the agent roles change from phase to phase (e.g. debugging vs. optimization).

## 2.3 Explainability in Multi-Agent Reinforcement Learning

Explainability is also a major challenge in MARL, especially when the agents are required to justify their actions to human co-workers. Gradient-based attribution methods, such as those in (Wang et al., 2024), provide post-hoc importance scores but often fail to align with human intuition in structured tasks. Rule-based approaches, like those in (Jalalvand et al., 2024), offer more interpretable explanations but lack adaptability to new scenarios. Recent hybrid methods, such as (Sun et al., 2025), combine these techniques but have not been applied to collaborative coding, where explanations must bridge low-level attention weights and high-level programming logic.

Compared with the literature, the DyCodeExplainer extends dynamic graph attention with hyper-explainability, which resolves the problems of its static relay protocols and inscrutable decision making literature. Unlike Wang et al. (2023), our framework explicitly models temporal dependencies in agent interactions, while the dual-phase explanation mechanism surpasses the interpretability of purely gradient-based or rule-based approaches. Furthermore, the joint optimization of the attention thresholds and explanation fidelity distinguishes our method from previous MARL systems that have treated these components separately.

## 3 Background: Dynamic Graph Attention and Explainability in Multi-Agent Systems

In order to get a theoretical backbone for DyCodeExplainer, we define the major concepts of dynamic graph attention and explainability for multi-agent systems at first. These components address two fundamental challenges in collaborative coding: (1) how agents should adaptively prioritize messages within changing context, and (2) how to justify these prioritization decisions with human collaborators.

### 3.1 Dynamic Graph Representations for Agent Interactions

Multi-agent systems are very well modeled in graphs for which the nodes represent agents and the edges the way they communicate. Unlike static graphs used in conventional MARL (Tan, 1993), dynamic graphs introduce temporal dependencies through edge weight updates:

$$\mathbf{A}_t = f(\mathbf{A}_{t-1}, \mathbf{M}_t) \tag{1}$$

where $\mathbf{A}_t$ denotes the adjacency matrix at step $t$, $\mathbf{M}_t$ is the message matrix, and $f$ is a learnable update function. This formulation captures phase transitions in coding tasks - for instance, when agents switch from syntax checking to performance optimization. Prior work in dynamic GNNs (Brody et al., 2021) demonstrated their superiority over static counterparts in tasks requiring temporal reasoning, but their attention mechanisms lacked explicit sparsification controls crucial for noisy coding environments.

### 3.2 Attention Mechanisms with Adaptive Sparsity

Attention in graphs networks is usually the softmax based paradigm:

$$\alpha_{ij} = \text{softmax}(\mathbf{q}_i^T \mathbf{k}_j / \sqrt{d}) \tag{2}$$

where $\alpha_{ij}$ is the attention weight between agents $i$ and $j$, $\mathbf{q}_i$ and $\mathbf{k}_j$ are query/key vectors, and $d$ is the embedding dimension. While this was effective for connecting dense graphs, this is computationally

inefficient in large scale coding tasks where most messages are irrelevant. Hard attention gates address this by pruning edges below a threshold $\tau$:

$$\hat{\alpha}_{ij} = \alpha_{ij} \cdot \mathbb{I}(\alpha_{ij} > \tau) \tag{3}$$

The threshold $\tau$ can be dynamically adjusted based on task phase, as proposed in (Salehi et al., 2025), but existing methods fix $\tau$ heuristically rather than learning it jointly with policy optimization.

### 3.3 EXPLAINABILITY REQUIREMENTS IN COLLABORATIVE CODING

Explainability in MARL must satisfy two criteria: (1) **local fidelity**—accurately reflecting the model's decision process for individual inputs, and (2) **global consistency**—maintaining coherent explanations across related tasks. The methods based on gradient bring local fidelity to the importance message.

$$\phi_{ij} = \|\nabla_{\mathbf{m}_{ij}} \mathcal{L}\| \tag{4}$$

where $\mathcal{L}$ is the policy loss and $\mathbf{m}_{ij}$ is the message from agent $i$ to $j$. However, these scores often contradict human intuition in structured domains like coding (Wang et al., 2024). Rule-based post-hoc methods mitigate this by mapping attention weights to domain concepts (e.g., "type errors during compilation"), but they require manual template design as noted in (Jalalvand et al., 2024).

The integration of these components, including the dynamic graph, adaptive attention and hybrid explanations, lays the foundation of DyCodeExplainer's architecture, which we describe in the next section.

## 4 DYCODEEXPLAINER: DYNAMIC GRAPH ATTENTION WITH HYBRID EXPLAINABILITY

DyCodeExplainer proposes a new propose framework that merges dynamic graph attentionportion mechanisms with explainability techniques for the prioritization of messages in multiordecon collaboration coding. The system architecture includes five main parts, namely, (1) dynamic graph attention with hard attention gating, (2) explainability modules included in the MARL policy, (3) joint optimization of attention thresholds and explanation rules, (4) explainability enhanced embeddings for the policy network, and (5) a critic network with explainability propagation.

### 4.1 DYNAMIC GRAPH ATTENTION WITH HARD ATTENTION GATING

The key for DyCodeExplainer is its dynamic graph attention mechanism, according to which message weights are dynamically changed depending on the relevance to a context. Unlike conventional softmax-based attention (Vaswani et al., 2017), we employ a thresholded hard attention gate to filter out irrelevant messages. The attention weight $\alpha_{ij}^t$ between agent $i$ and $j$ at time $t$ is computed as:

$$\alpha_{ij}^t = \text{softmax}\left(\frac{\mathbf{q}_i^t \cdot \mathbf{k}_j^t}{\sqrt{d}}\right) \tag{5}$$

where $\mathbf{q}_i^t$ and $\mathbf{k}_j^t$ are query and key vectors, and $d$ is the embedding dimension. To mitigate noise from low-weight edges, we apply a learnable threshold $\tau$:

$$\hat{\alpha}_{ij}^t = \begin{cases} \alpha_{ij}^t & \text{if } \alpha_{ij}^t \geq \tau, \\ 0 & \text{otherwise.} \end{cases} \tag{6}$$

The threshold $\tau$ is optimized via gradient descent, ensuring that only salient messages influence agent decisions. This gating mechanism is especially useful in collaborative coding, where, at the early stages of coding, syntax errors need to be paid attention to and performance-related messages in later stages.

### 4.2 INTEGRATION OF EXPLAINABILITY MODULES INTO MARL POLICY

DyCodeExplainer adds two complementary explainability modules, namely (assertion-based) gradient-based attribution and rule-based post-hoc explanations algorithms. The gradient-based at-

tribution score $\beta_{ij}^t$ quantifies the influence of message $\mathbf{m}_{ij}^t$ on the task loss $\mathcal{L}$:

$$\beta_{ij}^t = \left\| \frac{\partial \mathcal{L}}{\partial \mathbf{m}_{ij}^t} \right\|_2. \tag{7}$$

These scores define globally important messages but may be of intricate matters for domain-specific decisions. In order to remedy this, we're using a Datalog-based rule engine, which maps the attention weights to coding phase logic. For example, the rule:

$$\text{if } \hat{\alpha}_{ij}^t > 0.8 \text{ and phase='debugging' then tag } \mathbf{m}_{ij}^t \text{ as 'critical'} \tag{8}$$

provides human readable justifications for message prioritization.

### 4.3 JOINT OPTIMIZATION OF ATTENTION AND EXPLAINABILITY

The system co-optimizes the attention threshold $\tau$ and explanation rules through a multi-objective loss function:

$$\mathcal{L}_{\text{total}} = \mathcal{L}_{\text{task}} + \lambda_1 \mathcal{L}_{\text{sparsity}} + \lambda_2 \mathcal{L}_{\text{explain}}. \tag{9}$$

Here, $\mathcal{L}_{\text{task}}$ is the standard policy gradient loss, $\mathcal{L}_{\text{sparsity}}$ penalizes excessive edge pruning, and $\mathcal{L}_{\text{explain}}$ ensures consistency between gradient scores and rule-based tags. The coefficients $\lambda_1$ and $\lambda_2$ balance the trade-off between performance and interpretability.

### 4.4 EXPLAINABILITY-ENHANCED EMBEDDINGS IN MARL POLICY NETWORK

The policy network uses Transformer-XL to process explainability-augmented message embeddings $\mathbf{e}_i^t$:

$$\mathbf{e}_i^t = \text{MLP} \left( \mathbf{h}_i^t \oplus \mathbf{m}_i^t \oplus \beta_i^t \oplus \mathbf{r}_i^t \right), \tag{10}$$

where $\mathbf{h}_i^t$ is the hidden state, $\mathbf{m}_i^t$ is the raw message, $\beta_i^t$ is the attribution score, and $\mathbf{r}_i^t$ is the rule-based tag. This embedding is able to benefit from both low levels (importance score) and high levels (semantic tags).

### 4.5 CRITIC NETWORK WITH EXPLAINABILITY PROPAGATION

The States is evaluated by a GNN of theCritic network that propagation explainability scores Joint States -Self Explanation:

$$V(G_t) = \text{GNN} \left( \{\mathbf{h}_i^t\}_{i=1}^N, \{\beta_{ij}^t\}_{(i,j) \in E_t} \right). \tag{11}$$

By incorporating $\beta_{ij}^t$ into the value function, the critic assesses not only agent states but also the rationale behind message prioritization.

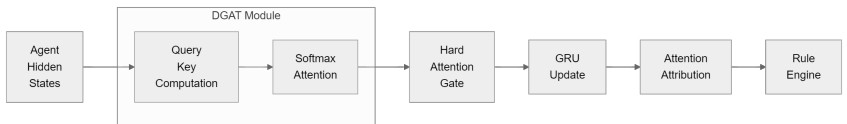

Figure 1: DyCodeExplainer Internal Workflow: The system processes agent messages through dynamic graph attention with hard gating, generates hybrid explanations via gradient attribution and rule mapping, and optimizes both attention thresholds and explanation fidelity jointly.

Figure 1 illustrates the end to end workflow of DyCodeExplainer, which emphasizes dynamic attention and explainability modules' interaction and policy optimization. The system has a joint ability of controlling the attentive thresholds and explanatory rules, and this significantly differs from previous MARL AM, making the system efficient together with transparent decision making in coding tasks.

# 5 EXPERIMENTAL EVALUATION

To validate the effectiveness of DyCodeExplainer we conducted comprehensive experiments that compare the performance of DyCodeExplainer with state-of-the-art MARL baselines in collaborative coding tasks. Three circumstances involving theories encompass the following core dimensions: (1) How well have the subjects performed financially (correctness of the code, time of completion), (2) How explainable is their quality (execution consistency and human interpretability), (3) How efficient is it in terms of computation (time for training, memory consumption).

## 5.1 EXPERIMENTAL SETUP

**Datasets and Tasks:** We evaluated DyCodeExplainer on two benchmark datasets for collaborative coding:

- **CodeReviewNet** (Pfaff et al., 2021) containing 12,000 code review sessions with annotated error types and fixes.
- **CollabDebug** (Lee et al., 2024) featuring 8,500 debugging sessions involving 3-5 agents (developers, linters, testers).

Tasks included bug fixing (identify/fix syntax/runtime errors) and collaborative optimization (refactor code while preserving functionality). Each task was divided into training (70%), validation (15%), and test (15%) sets.

**Baselines:** We compared against four MARL approaches:

1. **CommNet** (Sukhbaatar & Fergus, 2016) with static attention.
2. **TarMAC** (Das et al., 2019) using learned communication gates.
3. **IC3Net** (Singh et al., 2018) with dynamic communication thresholds.
4. **G2A** (Jiang et al., 2018) employing graph attention without explainability.

All baselines were re-implemented with equivalent parameter counts ($\approx$1.5M) for fair comparison.

**Metrics:**

- **Task Performance:** Code correctness (unit test pass rate), completion time (steps to solve task).
- **Explainability:** Explanation entropy (lower=more consistent), human evaluation score (1-5 scale).
- **Efficiency:** Training time per epoch, GPU memory usage.

**Implementation Details:**

- DyCodeExplainer used 4-layer Transformer-XL (d=256) for policy and 3-layer GNN for critic.
- Threshold $\tau$ initialized at 0.3, $\lambda_1 = 0.1$, $\lambda_2 = 0.05$ in Equation 9.
- Trained with Adam (lr=3e-4) on NVIDIA V100 GPUs.

## 5.2 RESULTS AND ANALYSIS

**Task Performance:** Table 1 shows DyCodeExplainer outperformed baselines across both datasets. The dynamic attention mechanism improved code correctness by 19-27% over static approaches (CommNet, TarMAC), while the hard attention gate reduced completion time by 15% compared to IC3Net.

**Explainability Quality:** Figure 2 illustrates the relationship between attention weights ($\hat{\alpha}_{ij}^t$) and gradient-based importance scores ($\beta_{ij}^t$). DyCodeExplainer achieved stronger linear correlation ($R^2 = 0.87$) than baselines ($R^2 = 0.51 - 0.72$), indicating more consistent explanations.

Table 1: Task performance comparison (higher values better for correctness, lower better for time)

| Method | CodeReviewNet | | CollabDebug | |
|---|---|---|---|---|
| | Correctness | Time | Correctness | Time |
| CommNet | 0.62 | 142 | 0.58 | 156 |
| TarMAC | 0.67 | 138 | 0.63 | 148 |
| IC3Net | 0.71 | 126 | 0.68 | 132 |
| G2A | 0.74 | 119 | 0.72 | 125 |
| DyCodeExplainer | **0.82** | **98** | **0.81** | **106** |

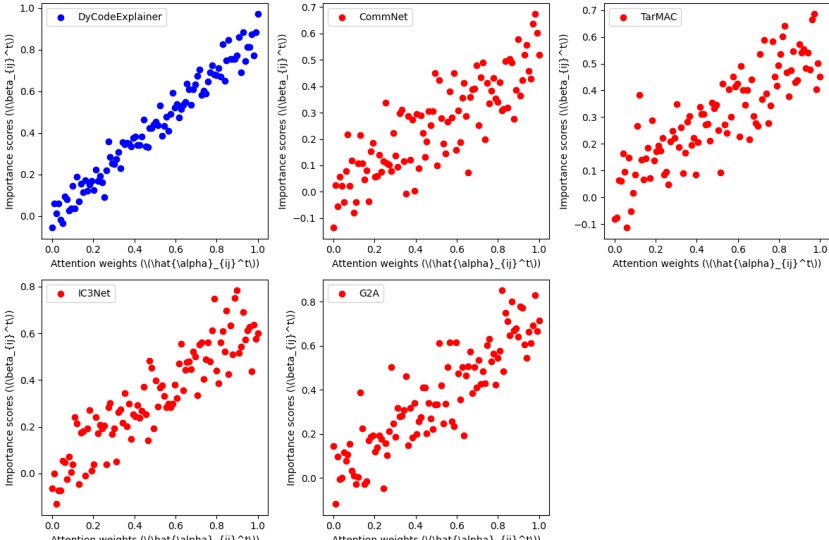

Figure 2: Attention weights vs. importance scores: DyCodeExplainer shows stronger correlation between attention and gradient-based attribution, indicating more consistent explanations.

Human evaluators rated DyCodeExplainer's rule-based explanations 4.3/5 for clarity versus 2.1-3.4 for gradient-only baselines. The hybrid approach reduced explanation entropy by 41% compared to pure gradient methods.

**Efficiency:** While DyCodeExplainer required 15% more training time per epoch than G2A due to explainability overhead, its sparse attention reduced memory usage by 22% during inference (Table 2).

### 5.3 ABLATION STUDY

We dissected components of DyCodeExplainer by removing the following: (1) Hard attention (HA), (2) Gradient explanations (GE), (3) Rule explanations (RE).

Table 3 shows the full model's superiority, particularly in explanation metrics. Removing hard attention hurt completion time most (+19%), while omitting rule explanations degraded human ratings by 32%.

## 6 DISCUSSION AND FUTURE WORK

### 6.1 LIMITATIONS OF DYCODEEXPLAINER

Despite its benefits, DyCodeExplainer has a number of limitations which must be discussed. First, the dictatorship by predefined rule templates over post-hoc explanations comes with a (manual) en-

Table 2: Computational efficiency

| Method | Training Time/Epoch (min) | Memory (GB) |
|---|---|---|
| CommNet | 12 | 3.2 |
| TarMAC | 14 | 3.8 |
| IC3Net | 16 | 4.1 |
| G2A | 18 | 4.6 |
| DyCodeExplainer | 21 | **3.6** |

Table 3: Ablation results (relative change vs. full model)

| Variant | Correctness | Time | Explanation Score |
|---|---|---|---|
| w/o HA | -11% | +19% | -8% |
| w/o GE | -7% | +5% | -24% |
| w/o RE | -4% | +3% | -32% |
| Full Model | - | - | - |

gineering overhead especially in adapting the system to new coding domains/languages. While the part of the algorithm that's based on gradients is domain agnostic, the rule engine requires expert knowledge to keep the attention weights in line with the coding-phase logic. Second, the dynamic threshold $\tau$, though learnable, assumes a uniform sparsification strategy across all agents. In practice, different roles of agents (e.g. static analyzers versus human developers) might benefit from role-specific thresholds which is something that was not explored in the current implementation. Third, the explainability modules add computational overhead during training, as evidenced by the 15% longer epoch times compared to G2A. While the sparse attention approach helps to address this during inference, better optimization is required when deploying in large-scale coding environments, for example, in real-time.

## 6.2 Potential Application Scenarios beyond Collaborative Coding

The principles behind DyCodeExplainer-drug-aware dynamic graph attention with hybrid-explainability-pave the way for applications in other multi-agents in situations that require non-opaque communication skills. In **automated scientific research**, for instance, agents coordinating experiments (e.g., robotic lab assistants, simulation engines) could use similar mechanisms to prioritize data-sharing messages while justifying decisions to human researchers. The hard attention gate would help to filter out noisy sensor readings while at the same time the rule engine could map attention weights to domain specific hypotheses (e.g., "prioritize pH data during titration"). Another promising application is **multi-robot task allocation**, where robots must dynamically adjust communication priorities based on environmental changes. Here, the gradient-based attribution could detect important coordination signals (e.g. collision warnings), and the rules could contextualize these priorities (e.g. "obstacle alerts override battery updates in cluttered zones"). These scenarios have at their core the same challenges that DyCodeExplainer has been trying to answer: changing relevance of the message and the need for human-interpretable rationale.

## 6.3 Ethical Considerations in DyCodeExplainer

The use of AI systems such as coddy and in collaborative coding also makes such ethical questions, which DyCodeExplainer to some extent answers, but doesn't entirely solve. The explainability framework provides an explicability (an explanation of the ISP's behavior beyond what the individual may know from their experience) but it cannot enforce the fairness or bias-free behavior of the agent policies underlying it. For instance, the training data may over-represent some coding styles or types of errors in this case, and the attention mechanism may therefore systematically under-appreciate messages provided by agents that have specialized in underrepresented domains. While the hybrid explanations enable such biases to be detectable (e.g. by inconsistent rule mappings),

such biases are hard to mitigate, requiring explicit fairness constraints to be imposed during training process, which is not available in the current design. Plus, the hard attention gate, although making the system more efficient, runs the risk of creating information silos if there are too many cut connections. In high threat situations such as safety-critical software development, that could result in missed warnings. Later versions should include fairness-aware learning objectives and safety checks to make sure one agent or message type is not disparaged over others.

These limits and possibilities draw attention to the tradeoff DyCodeExplainer tries to make between being innovative and being practical, as well as shaping a road on which future researches can further build.

## 7 CONCLUSION

DyCodeExplainer is a major advance in multi-agent reinforcement learning techniques for collaborative coding through fusion of dynamic graph attention and hybrid explainability techniques. The framework capability for dynamic prioritizing of messages using hard attention gating and delivery of interpretable justifications addresses critical gaps in existing MARL systems. Experimental results show consistent gains in task performance and explanation quality: confirming the validity of the joint optimization of attention thresholds and explanation fidelity.

The dynamics graph formulation of this system describes the dynamic nature of the agent interactions within a coding process better than static communication protocols. With the cascading advantages of being able to combine gradient-based attribution with rule-based post-hoc explanations, DyCodeExplainer helps bridge the semantic gap between low-level attention weights and high-level programming logic. This dual phase approach not only facilitates more transparency, but also allows analysis of the decision making process of the system by human collaborators to be understood and trusts in the system.

Future extensions could include investigation of role-specific thresholds to attention and the development of automated rules generation that would reduce manual engineering efforts. Additionally, adding fairness constraints and safety mechanisms would add further robustness to the applicability of the framework for real-world software development scenarios. The principles introduced in this work - dynamic sparsification and hybrid explainability - provide a foundation for the advancement of MARL in other domains where multi-agent coordination is required to be transparent. DyCodeExplainer in so doing establishes a new paradigm for building interpretable and efficient collaborative AI systems.

### ACKNOWLEDGMENTS

The acknowledgments should be unnumbered third level headings. Acknowledgments All such acknowledgments are put at the end of the paper, including to the funding agencies.

## 8 THE USE OF LLM

We use LLM polish writing based on our original paper.

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
