# OpenReview forum: "DyCodeExplainer: Explainable Dynamic Graph Attention for Multi-Agent Reinforcement Learning in Collaborative Coding"
_ICLR.cc/2026/Conference — Submitted to ICLR 2026_

### Official Review · Reviewer_PFrr · 2025-10-23

**Soundness:** 2
**Presentation:** 3
**Contribution:** 2
**Rating:** 2
**Confidence:** 3

**Summary:**

This paper proposes **DyCodeExplainer**, a framework designed to enhance both the performance and interpretability of multi-agent reinforcement learning systems for collaborative programming. The model uses a **dynamic graph attention network** to model agent interactions, where edges are gated by a sparsity-inducing hard threshold. It also introduces a **hybrid explainability module** that combines gradient-based attribution with symbolic rule extraction, aiming to produce interpretable reasoning traces for each coding action. The experiments demonstrate improved correctness and communication efficiency on two collaborative coding benchmarks.

**Strengths:**

* Tackles an emerging and relevant problem—explainable multi-agent collaboration for code generation.

* The dynamic attention mechanism is conceptually straightforward and may encourage efficient communication.

* The combination of symbolic rules and neural explanations is an interesting hybrid approach that could inspire follow-up work.

* The paper is clearly structured and includes good qualitative visualizations of attention maps.

**Weaknesses:**

* The hard gating function is non-differentiable, yet the paper does not describe how gradients are approximated (e.g., STE, Gumbel-softmax).

* The rule learning process is underdefined—there is no description of the search space or optimization mechanism for symbolic rules.

* The human evaluation used to justify the explainability claims lacks methodological rigor (no inter-rater reliability, sample size, or blinding).

* Experimental validation is narrow (two datasets) and lacks cross-language or cross-agent generalization tests.

* The proposed approach, while creative, seems only partially implemented and evaluated.

**Questions:**

1. How is the threshold parameter for the hard gate trained or tuned?

2. How are the symbolic rules parameterized and optimized alongside the neural model?

3. Can you provide more methodological details on your human evaluation protocol?

4. Have you tried the method on different coding environments or programming languages?

---

### Official Review · Reviewer_mHmD · 2025-10-28

**Soundness:** 1
**Presentation:** 1
**Contribution:** 1
**Rating:** 0
**Confidence:** 5

**Summary:**

The paper presents DyCodeExplainer, a framework that uses dynamic graph attention and explainability techniques to improve multi-agent reinforcement learning for collaborative coding.

**Strengths:**

This paper appears to be AI-generated and lacks genuine research depth. It loosely applies MARL concepts to LLM-based agent collaboration without a clear problem definition or credible technical grounding. The proposed framework mixes unrelated components like Transformer-XL, GNN, and explainability modules in a way that feels incoherent and artificial. There are essentially **no meaningful strengths**, as the work lacks novelty, theoretical justification, and real experimental validity.

**Weaknesses:**

1. The paper mixes multi-agent reinforcement learning, dynamic graph attention, and explainability mechanisms without establishing any clear theoretical connection, making it appear as an arbitrary combination of unrelated concepts.
2. The collaborative coding task is not naturally suited for MARL modeling and aligns more closely with LLM-based agent systems. The use of a reinforcement learning framework feels forced and unjustified, giving the impression that the work was written by someone unfamiliar with both fields who attempted to merge them superficially.
3. Although the experimental data and results appear complete, they lack real significance and reproducibility. No implementation details or accessible code are provided, making the reported findings highly questionable.
4. The writing is excessively templated, repeatedly using terms such as “dynamic graph attention” and “explainability framework,” and lacks the logical flow expected in genuine academic writing.
5. The proposed joint optimization objective and explainability modules have no theoretical grounding or derivation; the equations are merely formal decorations without substance.
6. The so-called “collaborative coding” task is poorly defined, with no clear description of the environment or the concrete interaction mechanisms among agents.

Overall, this paper appears to be an AI-generated pseudo-academic text that is formally structured but substantively empty, lacking genuine research foundation, methodological rigor, and technical credibility.

**Questions:**

This paper is clearly an AI-generated and fabricated manuscript with empty content, incoherent logic, and no genuine experiments or theoretical grounding. The text shows extensive signs of patching and template-based generation, with irrelevant citations, mismatched methods and tasks, and even indications of fabricated experimental data and results. It wastes the reviewers’ time and undermines the seriousness and integrity of the academic review process. It is strongly recommended that the conference committee verify the authors’ identities and the source of the submission and hold them accountable for this misconduct.

---

### Official Review · Reviewer_k1wM · 2025-10-29

**Soundness:** 2
**Presentation:** 3
**Contribution:** 2
**Rating:** 6
**Confidence:** 2

**Summary:**

This paper presents DyCodeExplainer, a novel multi-agent reinforcement learning (MARL) framework that integrates dynamic graph attention with explainability techniques to enhance collaborative coding. The idea is innovative and addresses critical challenges in message prioritization and decision-making transparency within collaborative coding environments.

**Strengths:**

The integration of dynamic graph attention networks (DGAT) with hybrid explainability techniques is a novel approach that effectively captures the evolving nature of agent interactions in collaborative coding tasks.

The combination of gradient-based attribution and rule-based post-hoc explanations provides both precise importance scoring and intuitive rationale generation, addressing the semantic gap between low-level attention weights and high-level coding logic.

**Weaknesses:**

The dependency on predefined rule templates for post-hoc explanations requires manual engineering effort, which may limit scalability and adaptability to new coding domains or languages.

**Questions:**

No

---

### Official Review · Reviewer_YmaW · 2025-10-29

**Soundness:** 2
**Presentation:** 1
**Contribution:** 1
**Rating:** 2
**Confidence:** 4

**Summary:**

This paper proposes a computational framework to combine dynamic graph attention and explainability to improve multi-agent performance in a collaborative coding task. The authors evaluate their proposed method with baseline MARL methods which indicate that DyCodeExplainer achieves better performance in terms of correctness and efficiency.

**Strengths:**

The idea of combineing dynamic graph attention and explainability in MARL is relatively novel.

**Weaknesses:**

The literature review section misses important work in the field when identifying the gap. In Section 2.3 the references are survey papers. I would recommend citing the exact empirical work. Attached are a few references to start with. There is work using an attention mechanism [6] or gating function [1-2] to selectively communicate given the task context. In the emergent communication community, researchers have also been working on improving communication interpretability by rewarding agents to communicate in a semantically meaningful space [5] or directly aligning the agent communication space with the human natural language space [3-4].

[1] Learning when to Communicate at Scale in Multiagent Cooperative and Competitive Tasks

[2] Interpretable learned emergent communication for human-agent teams

[3] Multi-agent cooperation and the emergence of (natural) language

[4] Language grounded multi-agent reinforcement learning with human-interpretable communication

[5] Emergent discrete communication in semantic spaces

[6] Multi-agent graph-attention communication and teaming

The target task “collaborative coding” is not explained before being used. I would recommend properly defining the task space (number of agents, form of communication, observation and action space) and the RL learning objective in a separate problem formulation section.

The proposed framework is not clearly explained in the method section. Details are missing to reproduce the work. Figure 1 should be extended to include interaction with external task environments. It would be helpful to provide a concrete interaction example between agents in the target task context.

Human evaluation details are missing. For example, how many annotators were recruited? What instructions and explanation materials were shown to participants for the evaluation? Are the reported differences statistically significant?

It's not clear to me why adding the explainability objective improves performance in the ablation study. My understanding is that explanations are generated for humans and are basically an auxiliary task that does not directly contribute to the main task objective. Could the author elaborate more on this?

**Questions:**

It's not clear to me why adding the explainability objective improves performance in the ablation study. My understanding is that explanations are generated for humans and are basically an auxiliary task that does not directly contribute to the main task objective. Could the author elaborate more on this?

---

### Meta-Review · Area_Chair_z2Jp · 2025-12-30

**Summary:**

The paper proposes DyCodeExplainer, a MARL framework integrating dynamic graph attention (DGAT) and explainability techniques for collaborative coding tasks. The primary concerns from the reviewers center on methodological rigor, novelty justification, and reproducibility.

**Reviewer Concerns:**

Reviewer YmaW identified critical concerns: (1) insufficient literature review failing to position the work against key empirical studies on attention/gating in MARL communication; (2) undefined "collaborative coding" task space (agents, communication protocols, RL objective); (3) opaque methodology lacking implementation details and environmental interaction context; and (4) unvalidated human evaluation. The authors provide no response to these comments.

**Reviewer Scores:**

This paper received four review comments, with scores of 2/6/0/2 respectively. As the score is far below the bar of ICLR, and the authors provide no response, it is obvious that this paper should be rejected.

---

### Decision · Program_Chairs · 2026-01-26

Reject